# West Nile, Sindbis and Usutu Viruses: Evidence of Circulation in Mosquitoes and Horses in Tunisia

**DOI:** 10.3390/pathogens12030360

**Published:** 2023-02-21

**Authors:** Youmna M’ghirbi, Laurence Mousson, Sara Moutailler, Sylvie Lecollinet, Rayane Amaral, Cécile Beck, Hajer Aounallah, Meriem Amara, Ahmed Chabchoub, Adel Rhim, Anna-Bella Failloux, Ali Bouattour

**Affiliations:** 1Laboratoire Des Virus, Vecteurs et Hôtes (LR20IPT02), Institut Pasteur de Tunis, Université Tunis El Manar, Tunis 1002, Tunisia; 2Institut Pasteur, Department of Virology, Arboviruses and Insect Vectors, 25-28 Rue du Docteur Roux, 75724 Paris, France; 3UMR BIPAR, Animal Health Laboratory, INRAE, ANSES, Ecole Nationale Vétérinaire d’Alfort, Université Paris-Est, 94704 Maisons-Alfort, France; 4ANSES, INRAE, Ecole Nationale Vétérinaire d’Alfort, UMR VIROLOGIE, Laboratoire de Santé Animale, 94700 Maisons-Alfort, France; 5National School of Veterinary Medicine, Sidi Thabet, University of Manouba, La Manouba 2010, Tunisia

**Keywords:** mosquito-borne-viruses, horses, high-throughput real-time PCR, microsphere immunoassay, seroneutralization test, cELISA assay

## Abstract

Mosquito-borne diseases have a significant impact on humans and animals and this impact is exacerbated by environmental changes. However, in Tunisia, surveillance of the West Nile virus (WNV) is based solely on the surveillance of human neuroinvasive infections and no study has reported mosquito-borne viruses (MBVs), nor has there been any thorough serological investigation of anti-MBV antibodies in horses. This study therefore sought to investigate the presence of MBVs in Tunisia. Among tested mosquito pools, infections by WNV, Usutu virus (USUV), and Sindbis virus (SINV) were identified in *Cx. perexiguus*. The serosurvey showed that 146 of 369 surveyed horses were positive for flavivirus antibodies using the cELISA test. The microsphere immunoassay (MIA) showed that 74 of 104 flavivirus cELISA-positive horses were positive for WNV, 8 were positive for USUV, 7 were positive for undetermined flaviviruses, and 2 were positive for tick-borne encephalitis virus (TBEV). Virus neutralization tests and MIA results correlated well. This study is the first to report the detection of WNV, USUV and SINV in *Cx. perexiguus* in Tunisia. Besides, it has shown that there is a significant circulation of WNV and USUV among horses, which is likely to cause future sporadic outbreaks. An integrated arbovirus surveillance system that includes entomological surveillance as an early alert system is of major epidemiological importance.

## 1. Introduction

Over the past 50 years, the significant increase in emerging and epidemic vector-borne diseases has changed perceptions about their impact on global mortality and their implication for socioeconomic and public health [1]. Today, over 500 arthropod-borne viruses are recognized worldwide, and at least 150 species are implicated in human, animal, or zoonotic diseases [2,3]. In the Mediterranean area, several arboviruses have been discovered in the last few decades [2,4]. In addition to *Aedes* mosquitoes, mosquitoes of the genus *Culex* (~768 taxa) include the most ubiquitous and important vectors of human and animal zoonotic pathogens. In the current situation where environmental and climatic changes are affecting the geographic range of these mosquitoes, *Culex*-borne pathogens are of particular concern. Human pathogens vectored by *Culex* mosquitoes currently include flaviviruses (West Nile virus (WNV), the Usutu virus (USUV), Japanese encephalitis virus, the St. Louis encephalitis virus) and alphaviruses (Western equine encephalomyelitis virus and Venezuelan equine encephalomyelitis virus) [5].

West Nile virus, maintained in enzootic transmission cycles between birds and *Culex* mosquitoes, is currently recognized as one of the most widely circulating and prevalent encephalitic flaviviruses [6], including in the Mediterranean region [7], that affects mainly humans and horses, which are considered to be dead-end hosts [8,9].

In North Africa, WNV has been circulating actively for decades. Its geographical position on migratory bird routes and a climate and environment that are favorable to *Culex* vectors (*Cx. pipiens* and *Cx. perexiguus*) have caused the virus to reemerge recurrently and unpredictably [10,11,12,13,14]. Several studies have shown the intensive circulation of WNV between mosquitoes and birds with a spillover to humans and equines. In humans, WNV provoked an outbreak in Tunisian’s central coastal region first in 1997 (111 cases, 8 deaths) [15], then in 2003 (112 cases, 9 deaths), again in 2012 (86 cases, 12 deaths), and again in 2018 (53 cases, 2 deaths). Citizens in different regions of the country were affected. In addition, sporadic cases were reported in 2007, 2010, and 2011 [12,15,16] mainly in the coastal areas of central regions in Tunisia but also in southern areas around oases. Clinical and serological studies have proven the prevalence of the infection of equines and birds with WNV mainly in northeastern governorates, along the eastern coast, and in the lowlands of southern Tunisia [16,17,18,19,20,21,22,23]. In neighboring Algeria, cases of human meningoencephalitis cases related to WNV infections were reported in 1994, 2012, and between 2013 and 2014 [24]. Recently there has been a significant seroprevalence of WNV antibodies in wild birds [25]. In several regions of Morocco, many studies have shown that WNV is circulating in humans, horses, birds, and mosquitoes [26,27,28,29].

Usutu virus (USUV, *Flavivirus*, *Flaviviridae*), was first isolated in South Africa in 1959 [30]. USUV shares many features with WNV: both are phylogenetically close and share a similar ecology; co-circulation is frequently observed in nature [31]. The clinical relevance of USUV as a human pathogen has been hypothesized since USUV-related infections were first reported in humans [32]. Human infection with USUV is most often asymptomatic or causes only mild clinical signs. Nonetheless, today, neuro-invasive cases of USUV have been reported in humans and vertebrates, and particularly in birds in Europe [33]. Past USUV exposure was found in horses [34] and migratory birds in Iran [35]. However, the effective role of USUV as a horse pathogen has yet to be clarified. In Tunisia, Ben Hassine et al. [36] reported the presence of anti-USUV antibodies in ten equines in southwest Tunisia. The virus was also found in the abundant laughing dove (*Spilopelia senegalensis*), a resident bird in Tunisian oases [37]. USUV was detected in *Culex pipiens* and *Cx. perexiguus* collected in Algeria and southern Spain [36,37,38]. USUV was found in other mosquito species such as *Cx. pipiens* [39], *Culiseta. annulata*, *Aedes albopictus*, *Ae. japonicus* [31], *Ochlerotatus detritus*, *Oc. caspius* and *Anopheles maculipennis s.l.* collected in Northern Italy [40,41,42,43]. Both viruses were found in native *Culex* mosquito species such as *Cx. modestus* and *Cx. perexiguus* in the marshlands of some southern European countries [44,45,46,47,48]. Different field-collected native (*Ae. vexans*) and invasive mosquitoes of the *Aedes* and *Culex* genera (*Ae. albopictus*, *Ae. detritus*, *Cx. torrentium*, *Ae. japonicus*) were proven to be able to transmit WNV experimentally [44,45,46,47,48].

Tick-borne encephalitis virus (TBEV) is the most important neuroinvasive arbovirus transmitted and endemic in many European countries [49]. There is scant information on TBEV epidemiology in Tunisia [50]. Tick-borne encephalitis has been detected in the EU since 2012 [51]. Hard ticks (Ixodidea), and particularly those of the genus *Ixodes*, are the main vectors of TBEV [52]

Unlike WNV, the Sindbis virus (SINV) is widely distributed across Europe and Africa and has been detected in mosquitoes and birds across Afro-Eurasia [53]. Human cases are reported mainly from South Africa, Finland, and Sweden [53]. SINV has been linked to neurological disease in horses in South Africa [54,55,56,57]. Ornithophilic *Culex* mosquitoes are the primary enzootic vectors of SINV in various geographical regions [54]. SINV has been isolated or detected in *Culex torrentium*, *Cx. pipiens*, Cx. theileri, Cx. perexiguus, Cx. univittatus, Culiseta morsitans, and *Aedes* mosquitoes [58].

The role of *Cx. perexiguus* as a vector for WNV was reported in Algeria, Portugal, Italy, and Spain even when found only in low abundances [36,42,58,59,60] and in Egypt, Senegal, and Israel [61]. Interestingly, in Spain, WNV seroprevalence in house sparrows, a resident wild bird species involved in WNV amplification in Europe and Northern Africa, was positively correlated with *Cx. perexiguus* abundance [62]. *Culex perexiguus* could play an important role in transmitting WNV in North African countries, where wetlands (rice fields, bodies of standing water) favor their breeding. *Cx. perexiguus* is mainly ornithophilic and feeds exclusively on birds early in the year but it can switch to larger mammals during the summer and autumn (WNV epidemics seasons) and can sometimes feed on humans [63]. This bridge behavior could foster the spillover of WNV to humans in areas shared by migratory birds and humans. Like the arboviruses mentioned above, SINV is also transmitted by mosquitoes, mainly of the genus *Culex*, and uses passerine birds as its natural host [64].

In Tunisia, despite the endemic circulation of WNV, no operational integrated surveillance program exists for arboviruses. The scarcity of molecular and serological data thwarts any epidemiological studies in the country. Obtaining baseline data on the distribution and the prevalence of arboviruses in vectors and hosts is therefore necessary for any regional and countrywide integrated surveillance. Consequently, we designed this study (i) to detect mosquito-borne viruses in Tunisia using a high-throughput tool based on the BioMark™ Dynamic matrix system in mosquitoes, and (ii) to evaluate the circulation of arboviruses in horses using a competitive enzyme-linked immunosorbent assay confirmed by Microsphere Immunoassays and microneutralization tests.

## 2. Materials and Methods

### 2.1. Experimental Design

High-throughput microfluidic real-time PCRs were performed to screen mosquitoes for arbovirus infections (flaviviruses and other arboviruses). Serological tools were used to evaluate past exposure of horses to flaviviruses. The experimental design is shown in Figure 1.

### 2.2. Study Location and Sample Collection

#### 2.2.1. Mosquito Sampling

In 2018 and 2019, adult mosquitoes were captured in four different localities (Ichkeul, Moknine, Monastir and Mseken) (Figure 2) where human cases of WNV had been recorded. Adult mosquitoes were collected using CDC miniature light traps (John W. Hock Co., Gainesville, FL, USA). Six CDC traps were placed in each sampling site, approximately 1.5 m above the ground. They were put near houses and animal shelters, and operated between 6:00 pm and 07:00 am each night, when the average temperature is around 22 °C. The traps were then transferred to the laboratory and kept in the freezer for 15 min to immobilize the mosquitoes. Species were identified with a binocular dissecting microscope and then examined using the software for the mosquitoes of Mediterranean Africa [65].

#### 2.2.2. Sampling Horses for Sera

Between 2018 and 2019, we also sampled 369 horses from 11 different localities located in 2 different bioclimatic zones, including the 4 localities operating the mosquito collection (Figure 2). In each region, the horses were randomly selected based on the availability of horse owners and the procurement of informed consent. The age, sex, and breed of each horse were recorded. All studied horses were born in Tunisia and none had traveled outside the country. Five milliliters of peripheral blood were collected from the jugular vein in tubes with no coagulant. The blood samples were centrifuged at 2500 rpm for 15 min and the sera were removed and stored at −20 °C for the serological tests.

### 2.3. Sample Processing and RNA Extraction

Identified mosquitoes were dissected above a freeze pack. Pools of ten abdomens of the same species were grouped and the remaining body parts (RBP; head and thorax) were stored individually at −80 °C to confirm mosquito infection. Some pools were composed of whole mosquitoes. The pools were ground in 500 µL of Dulbecco’s Modified Eagle’s Minimum Essential Medium (DMEM) supplemented with antibiotic solution and the homogenate was clarified by centrifugation at 6000× *g* for 2 min. RNA was extracted from 150 µL of each abdomen pool homogenate using Nucleospin RNA kit (Macherey-Nagel, Germany) as per the manufacturer’s instructions. Total RNAs were eluted in 60 μL of RNase free water and stored at −80 °C until the partial amplification of arboviruses.

The RBP (head/thorax) of individual mosquitoes were homogenized in 300 µL of DMEM with 10% fetal bovine serum using the homogenizer Precellys^®^ 24 Dual (Bertin, France) at 5500 rpm for 20 s. Total RNAs were extracted from 100 µL of homogenates using the Nucleospin RNA II extract kit (Macherey-Nagel, Germany). Total RNA per sample was eluted in 50 µL of RNase free water and stored at −80 °C until use.

### 2.4. Reverse Transcription and cDNA Pre-Amplification

To transcribe RNAs to cDNA, reverse transcription was conducted using the qScript cDNA Supermix kit according to the manufacturer’s instructions (Quanta Biosciences, Beverly, MA, USA). cDNA was then pre-amplified using the Perfecta Preamp Supermix (Quanta Biosciences, Beverly, MA, USA) according to Moutailler et al. [66]. Pre-amplified 1:5 diluted cDNAs were stored at −20 °C until use.

### 2.5. High-Throughput Real-Time PCR Screening

For the epidemiological surveillance of arboviruses, high-throughput microfluidic real-time PCR amplifications were performed according to Moutailler et al. [66] using BioMark™ real-time PCR system (Fluidigm, South San Francisco, CA, USA) and 96.96 dynamic arrays to perform 9216 individual reactions in one run. Sets of primers and probes used in this study are available [66].

Three controls were included in each dynamic array chip: (i) a negative water control to exclude contamination, (ii) a positive control with cDNA (virus reference material) or plasmid DNA, and (iii) an internal control to exclude PCR inhibitors (*Escherichia coli* DNA strain EDL933 using specific primers and probes targeting the *E. coli eae* gene) [67]. The data that were acquired were analyzed using the Fluidigm real-time PCR Analysis Software (Fluidigm, South San Francisco, CA, USA) according to [68].

### 2.6. Confirmation of a Disseminated Infection by Real-Time PCR of RBP

WNV positive abdomen mosquito pools were confirmed by WNV real-time PCRs on a One Step Instrument (Applied Biosystem, Thermo Fisher Scientific, Illkirch, France) by screening the cDNAs of RBP (head/thorax) of individual mosquitoes comprising each pool. The real-time RT-PCR assay targeting part of 5′UTR and capsid C regions [69] was performed in a final volume of 25 μL using 5 μL of RNA and AgPath-ID™One-Step RT-PCR Reagents (Thermo Fisher Scientific, France), with primers and probes at 400 nM and 200 nM, respectively. Thermal cycling conditions were as follows: 45 °C for 10 min, 95 °C for 10 min, 45 cycles at 95 °C for 15 s and 60 °C for 1 min. The positive SINV pools were further screened by SINV real-time RT-PCRs, as described by Moutailler et al. [66]. No validation was performed for positive whole mosquito pools since no RBP were available.

### 2.7. Serological Investigation of Flavivirus Infections in Horses by ELISA Assay

To detect anti-flavivirus antibodies in horses, serum samples were tested using a commercially available competitive Enzyme Linked Immunosorbent Assay (c-ELISA), the ID Screen^®^ West Nile Competition multi-species ELISA kit (cELISA; Innovative Diagnostics; Montpellier, France), that detects anti-E protein antibodies. The cELISA protocol was followed as per the manufacturer’s instructions. The OD for each sample was read at a wavelength of 450 nm. The results of the cELISA assays were calculated as a percentage of the negative control, as indicated by the manufacturer.

### 2.8. Microsphere Immunoassay

The cELISA positive serum samples were screened for WNV, USUV, and tick-borne encephalitis virus (TBEV) using a *Flavivirus* Microsphere immunoassay (MIA) as described by [70,71]. Briefly, recombinant soluble ectodomain of WNV envelope (E) glycoprotein (WNV.sE) and the recombinant E domains III (rEDIIIs) of WNV, USUV, and TBEV containing virus-specific epitopes were covalently bound to fluorescent beads. Reactivity against WNV.sE is indicative of the presence of anti-flavivirus antibodies and infections with WNV, TBEV and USUV could be discriminated using sera reactivity against WNV.EDIII, TBEV.EDIII, and USUV.EDIII. The cut-offs of WNV.sE, WNV.EDIII, and TBEV.EDIII antigens were found to be 17, 54, and 61, respectively, as described in [70]. For USUV, the cut-off was determined from the mean of median of fluorescence (MFI) values of the sera of 66 negative horses plus 3 standard deviations of the mean. The MIA results were interpreted as follows: (i) a serum was considered positive for WNV (or alternatively for USUV) if it reacted against WNV.sE and WNV.EDIII (or alternatively USUV.EDIII); (ii) in cases of positive reactions with several rEDIIIs for viruses belonging to the Japanese encephalitis serocomplex (i.e., USUV and WNV), an animal was considered infected with WNV (or alternatively for USUV) if the corresponding bead coupled to rEDIII generated an MFI that was at least twice greater than that generated with the other bead. The animal was considered infected with WNV or USUV if a two-fold difference could not be achieved; (iii) a serum was considered positive for TBEV when the TBEV.EDIII MFI was above the cutoff; and (iv) if the sample reacted with WNV.sE but not with any of the EDIIIs, it was considered positive for an undetermined *Flavivirus.*

### 2.9. Virus Neutralization Tests

*Flavivirus* cELISA-positive samples reacting with WNV.EDIII and USUV.EDIII in MIA were confirmed using virus-specific microneutralisation tests (MNT) against WNV and USUV. Neutralizing antibody titers against WNV and USUV were determined by MNT on Vero cells using WNV strain IS-98-ST1 (Genbank ID AF481864.1, provided by P. Desprès, IPP) and the USUV strain France 2018 (Genbank ID MT863562.1) as described by Beck et al. [70]. At least 90% protection of the cells should be obtained to consider a serum dilution as neutralizing. A serum was considered positive if cells were protected at the 1:10 serum dilution for WNV and USUV. Serum cytotoxicity was determined in serum control wells after the addition of serum dilutions and cells in the absence of virus supernatant. Besides TBEV, MNT was also performed on samples reacting with TBEV.EDIII in Vero cells with the TBEV Hypr strain (Genbank ID U39292.1). The virus with titers that were at least four times higher than the titers with the other two viruses was identified as the infecting *Flavivirus* [70].

### 2.10. Ethical Statement

The study was conducted according to the guidelines of the Declaration of Helsinki and was approved by the Commission on Ethics and Animal Welfare of the Institut Pasteur de Tunis, Tunisia (Protocol Code 2014/03/I/LR11IPT03/V1 and date of approval March 2014).

## 3. Results

### 3.1. High-Throughput Real-Time PCR Screening of Mosquito-Borne Viruses

A total of 2480 mosquitoes collected from the 4 studied localities (Ichkeul, Moknine, Monastir and Mseken) were analyzed using the high-throughput real-time PCR (Table 1).

The collection included 1590 mosquitoes in Ichkeul, 80 mosquitoes in Moknine, 470 mosquitoes in Monastir and 340 mosquitoes in Msaken. Four species were identified: *Aedes caspius*, *Ae. detritus*, *Culex perexiguus* and *Cx. pipiens*.

In the tested pools (*n* = 248), single infections by WNV (*n* = 10) and SINV (*n* = 3) were identified in *Cx. perexiguus* from Ichkeul. Additionally, mixed infections by SINV/WNV (*n* = 3) and USUV/WNV (*n* = 1) were detected in *Cx. perexiguus* pools from Ichkeul (Table 1).

### 3.2. WNV Disseminated Infections Confirmed in Culex perexiguus Sampled in Ichkeul

Disseminated WNV infection in *Culex perexiguus* mosquitoes was confirmed in three positive abdomen pools by screening the cDNAs of RBP (head/thorax) of the individual mosquitoes composing each pool with a WNV real-time PCR targeting a different fragment of the WNV genome than the one amplified in the Biomark array. Two RBPs from one WNV positive pool and one RBP from a WNV/SINV positive pool were confirmed to be WNV positive respectively, whereas ten RBP from the one SINV positive pool were found to be negative (Table 1). The confirmation for USUV could not be performed because the one positive pool for USUV and WNV was composed of whole *Cx. perexiguus* mosquitoes and not abdomens (Table 1).

### 3.3. Seroprevalence of Flavivirus, WNV and USUV, Infections in Horses

To detect anti-flavivirus antibodies in the sera of 369 horses, 3 techniques were used (i) cELISA detecting anti-flavivirus antibodies, (ii) the MIA test targeting three flaviviruses: WNV, USUV and TBEV, and (iii) the MNT (virus neutralization test) to confirm the presence of anti-WNV, anti-USUV and anti-TBEV antibodies. The seroprevalence by cELISA was 39.6% (146 from 369). Of these, only 104 were analyzed by the MIA and MNT serological methods, because of the small volumes of collected serum. The MIA test showed that 71.2% (74/104) of flavivirus cELISA-positive horses were positive for WNV, 7.7% (8/104) were positive for USUV, 6.7% (7/104) were positive for undetermined flaviviruses and 1.9% (2/104) were positive for TBEV (Table 2). In addition, the presence of antibodies against two flaviviruses was revealed in several equines: both WNV and TBEV (*n* = 1), USUV and WNV (*n* = 4) and USUV and TBEV (*n* = 1) (Table 2).

The MNT showed that 68.3% (71 of 104) were positive for WNV with titers between 20 and greater than 320, 7.8% (8 of 104) were positive for USUV with titers between 10 to 320 and 11.5% (12 of 104) were positive for WNV and/or USUV (Table 3). TBEV infection was not confirmed by MNT (Table 3).

The two methods gave comparable results for 74 MIA WNV positives (2 WNV-positive sera were found to be WNV MNT-negative). Conversely, 1 MIA-negative serum was found to be USUV-positive in MNT and 1 that was positive for undetermined flavivirus by MIA was found to be negative by MNT; 2 other sera could not be tested in MNT because of their elevated cell toxicity. All MIA USUV-positive sera were confirmed positive for USUV in MNT (Table 4).

## 4. Discussion

This study is the first to report the detection in Tunisia of the WNV, USUV, and Sindbis virus in *Culex perexiguus* mosquitoes, collected in Ichkeul, with a high-throughput tool based on the BioMark™ Dynamic matrix system in mosquitoes. In Tunisia, the first detection and isolation of WNV from mosquitoes was in *Culex pipiens* in 2014 [21]. The low WNV infection rate reported was consonant with other investigations: 0.15% for *Culex pipiens* in Morocco [27], 0.016–0.20% for *Cx. pipiens* in Germany [72], 1.2% for *Cx. pipiens* in Tunisia [18], 0.56% for *Cx. perexiguus* in Algeria [38], 0.24% for *Cx. interrogator*, 0.08% to 0.8% for *Cx. pipiens pipiens* and *Cx. theileri* in Iran [73,74] and 0.28% for *Cx. nigripalpus* in Chiapas, Mexico [75].

We also reported a seroprevalence in horses of WNV infection (18.9% by MNT test) that was higher than the seroprevalence of the USUV infection (2.2% by MNT) with an additional 12 WNV- and/or USUV-infected horses. In Tunisia, the circulation of WNV has been described since 1970 in humans [76]. Several studies have been conducted subsequently in mammals including equines, suggesting a low viral circulation [23,77,78]. The most important serological survey in equines was conducted in 2009 (1189 sera tested, 28%). It corroborated previous results, where infection rates were highest in regions with WNV outbreaks, including in the north-eastern governorates (Jendouba, 74%), the eastern coast (Monastir, 64%) and the lowlands of Chott El Jerid and Chott el Gharsa (Kebili, 58%, Tozeur, 52%) [79]. In addition, anti-WNV and anti-USUV antibodies were detected in 32% and 1% respectively of tested birds in Tunisian oases [80].

In our study, to improve the specificity of cELISA screening, *Flavivirus* positive samples were tested by MIA using WNV.EDIII, USUV.EDIII, and TBEV.EDIII virus-specific-epitopes antigens [70,81] and by MNT against WNV and USUV. Of the cELISA positive sera, only 104 of 146 were screened in MIA. More than half of the cELISA positive horses (79 of 146) were confirmed by MIA assays to be WNV positive, indicating that WNV is the major *flavivirus* circulating in the region. Similar results were reported in the equine population in Pakistan with 249 of 292 MNT confirmed WNV infections [82], and in Catalonia with 92 MNT-positive horses for WNV, 11 for USUV, 4 for TBEV, and 57 for undetermined flavivirus [83]. The seroprevalence rate is higher than that reported in endemic areas such as New York (2.6%) [84], northern Italy (2.08%) [85], Greece (1.5%) [86], and Turkey (14.9%) [87], and greater than that reported in non-endemic areas such as Lebanon [88], Jordan (The Hashimiah region) (8%) [89], and Iran (Mashhad region) (11%) [90].

Sera that were found to be positive for undetermined flaviviruses in MIA were confirmed by MNT as follows: one negative, two cytotoxic, two corresponding to WNV- and/or USUV-infected animals and two WNV-positive. These few discordant results could reflect the lower specificity of the *flavivirus* EDIII MIA technique. These two methods corroborated the detection of anti-flavivirus WNV and USUV antibodies in horses [70]. However, MIA TBEV positive sera were found to be negative when using TBEV MNT. Given this, ticks and vertebrate hosts such as horses infected with TBEV or a closely related tick-borne *flavivirus* warrant investigation. Interestingly, TBEV was recently reported in *Ixodes ricinus* ticks and in sheep in Tunisia [50,91].

We did not identify the *flavivirus* involved in 10.8–15.4% (16 of 104 in MIA, 11 of 102 in MNT) of positive cELISA horses. Of 102 anti-flavivirus positive ELISA sera, only 83 had specific WNV-neutralizing antibodies. Such negative reactions in MNT could correspond to low WNV antibody responses under the MNT detection threshold or to infections by other flaviviruses (e.g., yellow fever or Dengue viruses). Recently, cELISA-positive and WNV MNT-negative horses on French islands in the Pacific were reported to be infected by the Dengue and Zika virus [92]. Our serological results highlighted the importance of considering the risk of circulation of other zoonotic flaviviruses such as Dengue and Zika viruses, as *Ae. albopictus* was detected in Tunisia [93] and revealed to be experimentally competent to transmit these viruses [94].

In Tunisia, serological surveys of horses also showed that WNV circulates in horses in the north-east (74%), the eastern coast (64%) and the south-west (55%), and WNV and USUV cocirculate [36,79]. In contrast, the seroprevalence of WNV in different regions of Morocco was low, as reported using cELISA and MNT [28,29]. However, similarly high seroprevalences were reported in military horses (60%) and dogs (62%) in Morocco using cELISA [95]. This high rate of WNV seroprevalence was also reported in Algeria [36,96]. USUV and WNV have been shown to co-circulate and share the same vector species but no single infection by USUV was detected in mosquitoes, although it was shown to circulate previously in Tunisia and in Morocco [36,80]. Despite USUV RNA detection, further studies are needed to assess the status of USUV in RBP as it was not amplified by PCR. The lower detection in mosquitoes and the seroprevalence in horses suggest that USUV should be less prevalent than WNV in Tunisia, whereas the reverse is observed in several European countries such as France [31,97]. The high WNV-neutralizing titers of the positive sera are probably due to repeated exposure to the virus or to re-infections during the previous outbreak of 2018, confirming the intense circulation of the virus during this year in Tunisia. The absence of any outbreaks after 2018 despite the circulation of the virus could be attributed to the acquisition of a protective immunity and may result from the lack of entomological virus surveillance and of underreported asymptomatic or mild and/or non-specific symptomatic cases.

Our results confirm the active circulation of WNV in several areas of Tunisia since: (i) there was a high abundance of *Culex* mosquitoes that may play an important role in epizootic transmission to humans [98] (ii) the presence of migratory birds [99], and (iii) the ability of *Culex* to transmit WNV experimentally [10], and (iv) the presence of *Cx. perexiguus*, the most important ornithophilic mosquito species for WNV enzootic circulation in Europe and a significant bridge vector able to infect horses and humans [100]. In Tunisia, WNV disease outbreaks often occur among humans living in or near wetlands where high concentrations of birds come into contact with many ornithophilic mosquitoes [79] and lie along the major routes of migratory birds [19]. Ichkeul is characterized by a water reservoir that serves a resting stop for migratory birds, and is thus an ideal ecological niche for repeated contact between domestic or migratory birds and mosquitoes. This enables the virus to amplify in an enzootic cycle [11]. WNV has been responsible for sporadic outbreaks of the disease in humans and/or horses, between July and September in several Mediterranean countries [101,102], where the environmental factors and climatic conditions are favorable for maintaining the WNV transmission cycle [103].

This study confirms the active circulation in mosquitoes of WNV in Tunisia. Whereas we detected SINV and USUV RNA, we could not amplify them by RT-qPCR. This constitutes this study’s main limitation. Further studies are needed to confirm USUV- and SINV-disseminated infection in mosquitoes and their role as vectors for both viruses. Health authorities should establish and use an integrated early warning system to detect arboviruses in humans, animals and vectors, particularly, given environmental changes and more specifically global warming that favors vector development. In addition, citizen awareness should be raised as a preventive measure against known and unknown arboviruses infections.

## Figures and Tables

**Figure 1 pathogens-12-00360-f001:**
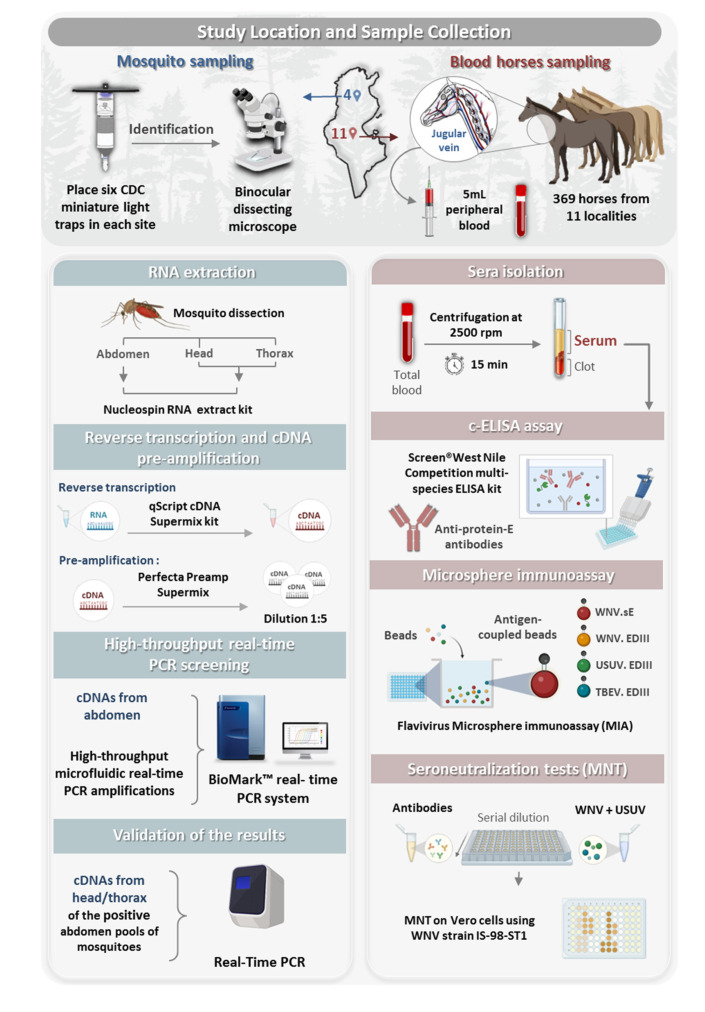
Schematic representation of the study workflow.

**Figure 2 pathogens-12-00360-f002:**
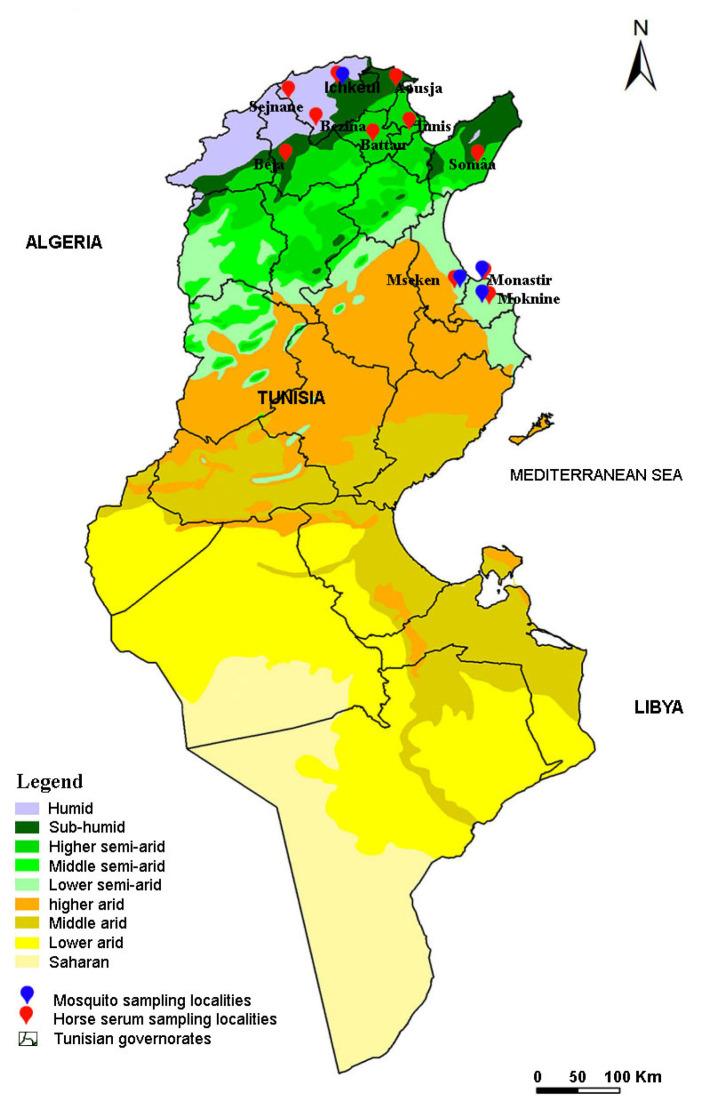
Map of Tunisia showing localities investigated for horse sera collection and mosquito sampling.

**Table 1 pathogens-12-00360-t001:** Arbovirus detection and infection confirmation of collected mosquitoes.

Localities (nb of Tested Pools, nb of Tested Mosquito Specimens) ^a^	Mosquito Species (nb of Tested Pools, nb of Mosquito Specimens)	Arboviruses Infection by High-Throughput Microfluidic Real-Time PCR (nb of Infected Pools Composed of 10 Abdomens + or 10 Whole Mosquitoes)	Arboviruses Infection Confirmation by Real-Time PCR (nb of RBP Confirmed/nb of RBP Tested) ^b^
Ichkeul (159, 1590)	*Ochlerotatus caspius* (12, 120)	Negative	n/a
	*Ochlerotatus detritus* (6, 60)	Negative	n/a
	*Culex perexiguus* (116, 1160)	SINV (1 + 2)	Negative (10/10)
		WNV (1 + 9)	WNV (2/10)
		WNV and SINV (1 + 2)	WNV and SINV (1/10)
		WNV and USUV (0 + 1)	n/a ^c^
	*Culex pipiens* (25, 250)	Negative	n/a
Monastir (47, 470)	*Oc. caspius* (16, 160)	Negative	n/a
	*Culex perexiguus* (23, 230)		n/a
	*Culex pipiens* (8, 80)		n/a
Moknine (8, 80)	*Oc. caspius* (3, 30)	Negative	n/a
	*Culex perexiguus* (4, 40)		n/a
	*Culex pipiens* (1, 10)		n/a
Mseken (34, 340)	*Oc. caspius* (24, 240)	Negative	n/a
	*Oc. detritus* (6, 60)		n/a
	*Culex perexiguus* (3, 30)		n/a
	*Culex pipiens* (1, 10)		n/a

^a^: mosquito specimens = abdomens or whole body; ^b^: two pools infected by SINV, nine pools infected by WNV, two pools infected by WNV and SINV and one pool infected by WNV and USUV were not confirmed because they were composed of ten whole mosquitoes rather than ten abdomens; WNV, West Nile virus; USUV, USUTU virus; SINV, Sindbis virus; nb: Number; ^c^: Not applicable, confirmation of RBP was done only for positive pools composed of ten abdomens and not for pools composed of ten whole mosquitoes.

**Table 2 pathogens-12-00360-t002:** Number of seropositive cELISA WNV horse sera and MIA confirmation of positive cELISA sera against Flavivirus, WNV, USUV and TBEV.

Localities (nb of Tested Sera by cELISA)	cELISA, WNV Results		MIA Analysis of Positive cELISA Horse Sera							
	Negative	Nb of cELISA Positive Sera/nb of Sera Analysed in MIA	Negative	Undetermined Flavivirus	WNV	USUV	TBEV	WNV + TBEV	USUV + TBEV	WNV + USUV
Aousja (15)	10	5/4	0	1	3	0	0	0	0	0
Battan (36)	26	10/9	3	0	3	3	0	0	0	0
Beja (11)	1	10/10	1	1	7	1	0	0	0	0
Bezina (2)	0	2/2	0	0	1	0	0	1	0	0
Ichkeul (38)	5	33/17	0	0	14	2	0	0	1	0
Moknine (32)	15	17/17	1	2	11	1	0	0	0	2
Monastir (32)	18	14/14	2	0	10	1	0	0	0	1
Mseken (22)	5	17/17	0	1	15	0	0	0	0	1
Sejnène (4)	0	4/4	0	0	4	0	0	0	0	0
Somâa (18)	12	6/2	0	0	2	0	0	0	0	0
Tunis (159)	131	28/8	2	2	4	0	0	0	0	0
Total (369)	223	146/104	9	7	74	8	0	1	1	4

**Table 3 pathogens-12-00360-t003:** Confirmation of positive cELISA horse sera using MNT against WNV and USUV.

Localities (nb cELISA Positive Sera/nb of Sera Analysed by MNT)	Negative	Undetermined *flavivirus*	WNV (Range of Titers)	USUV (Range of Titers)	TBEV	WNV and/or USUV
Aousja (5/4) ^a^	1	0	3 (40–80)	0	0	0
Battan (10/9)	3	0	3 (80–160)	3 (10–20)	0	0
Beja (10/10) ^b^	1	0	7 (20–≥320)	0	0	2
Bezina (2/2) ^c^	0	0	1 (160)	0	0	1
Ichkeul (33/17)	0	0	14 (20–160)	3 (40–320)	0	0
Moknine (17/17) ^d^	1	0	11 (40–≥320)	2 (20–80)	0	3
Monastir (14/14) ^e^	2	0	11 (20–≥320)	0	0	1
Mseken (17/17) ^f^	1	0	12 (20–≥320)	0	0	4
Sejnène (4/4)	0	0	3 (80–≥320)	0	0	1
Somâa (6/2)	0	0	2 (80–160)	0	0	0
Tunis (28/8) ^g^	2	0	4 (80–≥320)	0	0	0
Total (146/104)	11	0	71	8	0	12

^a^: One serum sample identified in MIA as positive for an undetermined flavivirus was found negative in WNV, USUV and TBEV MNT; ^b^: One serum sample identified in MIA as positive for an undetermined flavivirus and one serum found USUV-positive in MIA were found WNV- and/or USUV-positive in MNT; ^c^: One serum sample identified in MIA as WNV- and TBEV-positive was found WNV- and/or USUV-positive in MNT (identical neutralizing WNV and USUV antibody titers); ^d^:Two sera identified as flavivirus-positive in MIA were determined WNV-positive in MNT, one MIA-negative serum sample was determined USUV-positive in MNT and one serum sample identified in MIA as WNV-positive was found WNV- and/or USUV-positive in MNT; ^e^: One serum sample identified in MIA as WNV- and USUV-positive was determined WNV-positive in MNT and one serum identified in MIA as USUV-positive was determined as WNV- and/or USUV-positive in MNT; ^f^: Three sera identified as WNV-positive and one serum identified as flavivirus-positive in MIA were determined as WNV- and/or USUV-positive in MNT, one MIA WNV-positive serum was found MNT-negative, and one WNV/USUV-positive serum in MIA was found WNV-positive in MNT; ^g^: Two sera flavivirus-positive in MIA were cytotoxic and could not be tested in MNT.

**Table 4 pathogens-12-00360-t004:** Confirmation of positive cELISA horse sera using MNT and MIA tests against WNV and USUV.

Assay	Flavivirus Species Detected	nb of Positive Samples/nb of Tested Samples (%)
cELISA	*Flavivirus*	146/369 (39.5)
MIA	WNV	74/104 (71.2)
MNT	WNV	71/104 (68.3)
MIA	USUV	8/104 (7.7)
MNT	USUV	8/104 (7.7)

## Data Availability

Not applicable.

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
