# Peer review of "West Nile, Sindbis and Usutu Viruses: Evidence of Circulation in Mosquitoes and Horses in Tunisia"

_pathogens, 2023, doi:10.3390/pathogens12030360_

Round 1

Reviewer 1 Report

This manuscript is well written and has moderate novelty and interesting results. Please consider my comments as below:

-Lines 135 to 137: I suggest transfer to discussion.

-Line 177: Please explain the exact temperature.

-2.2.1 Mosquito sampling: Please let us know, how many mosquitoes (Culex, Anopheles, Aedes and ,...) have been captured in total and in each area?

Table 1: In front of Ichkeul, 5190 should be change to 1590?

Table 1: Please put (12,120) in front of Ochlerotatus caspius.

-Line 368 to 372: Please add and discuss the results of 2 related recent studies as below:

1- Shahhosseini, N.; Chinikar, S.; Moosa-Kazemi, S.H.; Sedaghat, M.M.; Kayedi, M.H.; Lühken, R.;Schmidt-Chanasit, J. West nile virus lineage—2 in culex specimens from Iran. Trop. Med. Int. Health 2017, 22, 1343–1349. 

2-Nariman Shahhosseini, Seyed Hassan Moosa-Kazemi , Mohammad Mehdi Sedaghat,Gary Wong, Sadegh Chinikar , Zahra Hajivand , Hamid Mokhayeri ,Norbert Nowotny  and Mohammad Hassan Kayedi ., Autochthonous Transmission of West Nile Virus by a New Vector in Iran, Vector-Host Interaction Modeling and Virulence Gene Determinants. Viruses, 2020, 12, 1449, 1-18.

-Discussion section: Authors should compare and discuss their results with  results of more other similar studies in Tunisia, Africa, Asia, Europe and America? 

Reviewer 2 Report

In this manuscript the authors (M’ghirbi et al.) describe their studies of arbovirus circulation in the most Northern part of Tunisia during 2018-19 using the combination of a multiplex qRT-PCR approach for detection of viral genomes in mosquitoes and a serological survey of horses – with some overlap of the sampling regions. The manuscript suffers from a string of short-comings that really hamper the reading and distracts from the essentials of the findings.

The Abstract is excessively long (~800 words). The MDPI Instructions to Authors stipulate the abstract should be a total of about 200 words maximum.

Similarly, the Introductions is excessively long, and then it does only focus on flaviviruses, yet the title of the manuscript includes an alphavirus (Sindbis virus (SINV)), which is not mentioned. There is also too much repetition and even irrelevant information in the Introduction.

The Discussion is also too long, with too much repetition and irrelevant comparisons to other parts of the world.

At the same time, there is not enough description of some of the methodologies used. For example, there are no details for the virus neutralization assay, including growth and titration of virus stocks and what cut off was used for neutralization (80% protection or some other level of inhibition of CPE?). And what does MNT stand for? It certainly cannot be an abbreviation of ‘seroneutralization tests’ – which in any circumstance should be “serum-neutralization test”. How was it determined that some sera were cytotoxic?

Why was only WNV results in the multiplex assay validated by RT-PCR – why not also Usutu and Sindbis viruses?

Why were sera screened for TBEV-specific antibodies when TBEV was not detected in the mosquitoes?

The manuscript would benefit from a complete rewrite also taking into account the following:

·       Correction of English grammar and syntax throughout (some examples given below).

·       Lines 47 and 446: please explain how an outbreak can be “foreseeable”

·       Lines 68-69: Why even introduce abbreviations for St Louis encephalitis virus, Western and Venezuelan equine encephalomyelitis virus, when they are not further dealt with in the manuscript. Moreover, the latter two viruses are alphaviruses, which probably should be pointed out to readers.

·       Line 88: the expression “circulation of WNV among humans, equines and birds” is misleading and could be misunderstood by readers unfamiliar with arboviruses. WNV circulates between mosquitoes and birds with spill-over to humans and equines. Humans and horses are not amplifying hosts and hence do not contribute to the “circulation”.

·       Table 1: there is no explanation for the abbreviation “nb”, but presumably it means ‘number’? Also, what does the star between numbers in the last column signify? The ‘a’ and ‘b’ should be superscript – otherwise they look more like spelling errors.

·       Table 3: just give the range of titers – not individual values – or the medium +SEM.

·       Lines 379-80: it is unclear why the mention of the possible lack of specificity of the cELISA is mentioned here, but not further discussed in terms of implication for the present study. In which case the sentence could be deleted.

·       Lines 403-405: there is o reference for the claim that horses on French pacific islands were infected with dengue and Zika virus, but it should presumably be PMID: 30730887?

·       Lines 416-417: this sentence does not make sense or is at best a non-sequitur.

·       Lines 422-23: no, the human outbreak does not explain the seroprevalence in horses, but both collectively certainly do point to high circulation intensity.

·       Line 426: what exactly is meant by ‘poorly symptomatic’? Is this mild, non-specific symptoms?

Minor points:

·       Line 116: correct to “(Spilopelia senegalensis), a resident bird ……..”

·       Line 172: correct to “mosquitos were captured”

·       Line 188: correct to “born in Tunisia and had not ….”

·       Line 287: correct to “Arbovirus detection….” In the table title.

·       Line 199: correct to “homogenate”.

·       Lines 199 & 205: inconsistency in how Macherey-Nagel is written (all capital letters in line 199). Please be consistent.

·       Line 203: correct to “fetal bovine serum”.

·       Line 215: delete ‘(2019)’.

·       Line 218: correct to “sets”

·       Line 306: correct to “369 horse sera” or “serum samples from 369 horses”

·       Line 311: correct to “serum volumes”

·       Line 323: correct to “greater than 320”

·       Line 334: correct to “one serum sample”. This also applies to the rest of the footnote for the table.

·       Line 360: correct to “in mammals including equines….. “

·       Line 362: correct to “survey in equines was conducted in 2009 (11 89 sera tested) and corroborated previous …..”.

·       Line 415: correct to “co-circulate”

·       Line 433: correct to “capable of infecting horses……”

END

Reviewer 4 Report

The authors present a study on the prevalence of flaviviruses and other mosquito-borne viruses in mosquitoes in Tunesia combined with a seroprevalence study of flavivirus infections in horses in Tunesia. For virus detection in mosquitoes, they use a high-throughput realtime PCR system (BioMark realtime PCR system developed by Moutailler et al., published in Viruses 2019) to screen for 64 arboviruses. They additionally use a confirmation West Nile Virus (WNV) realtime PCR to confirm (validate) positive results of the array, but do not use further confirmatory tests for other viruses (Usutu virus (USUV) or Sindbis Virus). Species determination was performed for 2480 mosquitoes trapped at 4 different localities in Tunesia, and 248 pools of 10 mosquito abdomens of the same species, each, were examined by the BioMark system. Positive reactivities were found for WNV in 10 pools, for Sindbis virus in 3 pools, for WNV+Sindbis in 3 pools and WNV+USUV in 1 pool, all for Culex perexiguus. Confirmatory realtime PCR using materials from the remaining bodies of the mosquitoes confirmed the occurrence of WNV RNA in 3 pools (line 299) or maybe in 3 mosquitoes belonging to 2 pools (line 302-303). This means that the BioMark results were confirmed by Realtime-PCR for WNV in only 2 or 3 out of 14 pools. It is unclear whether the USUV or Sindbis virus results of the BioMark system are of higher reliability, and why the authors did not perform confirmatory PCR assays for those viruses which seems to be necessary.

Regarding the seroprevalence study for flaviviruses in horses, the authors use a cELISA screening test for flaviviruses. The positive reactions were further investigated using a flavivirus -sphere microsphere immunoassay (MIA) able to discriminate among WNV, USUV and tick-borne encephalitis virus (TBEV) and a microneutralization assay (MNT) for confirmation of WNV and USUV. 146 of 369 horse sera originating from 11 different localities were positively reacting in the cELISA. A number of 104 out of 146 sera underwent investigation by MIA, and 71 % of the were shown to have WNV reactivity, while 7.7 %, 6.7 % and 1.9 % revealed reactivities for USUV, undetermined flaviviruses and TBEV, respectively. The MNT largely confirmed the reactivities for WNV and USUV.

Major remark:

The study presented here includes a large number of mosquitoes and horse sera and might increase the knowledge regarding arboviruses and flaviviruses in Northern Africa. Nevertheless, the results obtained by the BioMark system seem to be of low reliability as the authors show by the low conformity with the classical Realtime PCR assay performed for WNV only. To my opinion a confirmation of the other positive virus results by differing tests for USUV and Sindbis virus is necessary before the investigation can be published. Further, the results which have not been validated (confirmed) cannot be presented as prevalence values in the abstract and in the results.

Minor remarks:

The authors should shorten the introduction but nevertheless also include Sindbis virus (if detection was confirmed) and TBEV.

Round 2

Reviewer 2 Report

The authors are commended for having addressed a number of the concerns of this reviewer. However, there remains substantial issues with data presentation and interpretation. The Discussion is excessively long and disjointed, with much repetition of data from this study and disorganized iteration of published data. In general, the data does not substantially add to knowledge about arbovirus circulation in the Mediterranean region.

The manuscript could be further improved by also addressing the following:

·       Correction of English grammar and syntax throughout – there are still several examples of non-sequiturs and poorly phrased sentences that impedes understanding.

·       Line 22: what do the authors mean by “deep serological investigation”?

·       Line 26: correct to “positive for flavivirus antibodies”. Also define the abbreviation MIA on first usage.

·       Line 28: remove capital letters in tick-borne encephalitis virus – and introduce the abbreviation TBEV here (it is done in line 95).

·       Lines 53-54: change to “…. [6], including the Mediterranean region [7], affecting ……”

·       Lines 55-58: Break up the sentence running over four lines.

·       Line 65: correct to “central parts of Tunisia”

·       Line 73: correct to Usutu virus (USUV( was first ….”

·       Line 101: Besides,…….

·       Line 145: change to “mosquitoes. Identification of species was conducted using a binocular dissecting microscope ....”

·       Figure 2: change to “Horse serum sampling localities” in the legend in the actual figure.

·       Line 154: correct to “All studied horses were born ……”

·       Line 156: change to “Blood samples were …..”

·       Line 162: correct to “Pools were ground in 500 uL….”

·       Line 185: as a minimum provide the primers and probes in a supplementary table.

·       Line 203: correct to “as they did not have ….”

·       Line 208: correct to “The cELISA …”

·       Line 213: correct to “positive serum samples were…..”

·       Line 215: replace (2017) with [70]

·       Line 241: provide percentage cell protection  - presumably 90% according to the reply to reviewer – but state it in the manuscript too. Also provide information about how cell toxicity was assessed, since that is subsequently mentioned in both the Results section and in the Discussion.

·       Table 1: there is an extra b superscript.

·       Lines 263-5: sentence needs rephrasing in other to avoid the expression ‘among them’

·       Lines 301 & 302: correct to “found to be” in both sentences.

·       Table 3: either remove the left side border or include a right side border in the table.

·       Table 3: All the explanatory a, b, c, etc in the first column should be in superscript.

·       Table 4: for the cELISA it is not WNV that is detected but just flavivirus in general.

·       Lines 351-3: delete sentence starting with “of which 79 were …” as the same thing is said again below.

·       Lines 363-9: this is all recapitulation of data and should not be in the Discussion.

·       Line 373: correct to “a closely related ….”

·       Lines 378-9: please provide a reference for the claim that the flavivirus cELISA has low specificity.

·       The whole section in lines 376-406 is very disjointed and in need of editing/rewriting.

·       Line 412: correct to “capable of infecting ….”

·       Lines 413-415: sentence need rephrasing.

Author Response

Response to Reviewer 2 Comments

Point 1: The authors are commended for having addressed a number of the concerns of this reviewer. However, there remain substantial issues with data presentation and interpretation. The Discussion is excessively long and disjointed, with much repetition of data from this study and disorganized iteration of published data. In general, the data does not substantially add to knowledge about arbovirus circulation in the Mediterranean region.

Response 1:

Changes have been added throughout the text to make the text more consistent with major changes to the discussion, by shortening it and removing repetition data from this study and published data. The data reported in this study are believed to represent an important piece of a puzzle called arbovirus in the Mediterranean region as the data are reported for the first time in Tunisia.

Point 2: The manuscript could be further improved by also addressing the following: Correction of English grammar and syntax throughout – there are still several examples of non-sequiturs and poorly phrased sentences that impede understanding.

Response 2: As requested by the reviewer, we added some details to clarify the doubts and changed the manuscript accordingly. As suggested, in the second round of revision, the manuscript has been revised by a native English speaker.

Point 3: Line 22: what do the authors mean by “deep serological investigation”?

Response 3: To clarify the meaning, the sentence has been changed (Line 22: “Furthermore no study reported mosquito borne viruses (MBVs) or a thorough serological investigation of anti-MBV antibodies in horses”

Point 4: Line 26: correct to “positive for flavivirus antibodies”. Also define the abbreviation MIA on first usage.

Response 4: As requested, the phrase has been corrected (Line 26): “The serosurvey showed that 146/369 of horses were positive for flavivirus antibodies using cELISA test”.

Besides, MIA abbreviation has been defined in the abstract: “The microsphere immunoassay (MIA) showed that …”

Point 5: Line 28: remove capital letters in tick-borne encephalitis virus – and introduce the abbreviation TBEV here (it is done in line 95).

Response 5: Changes has been made accordingly (Line 28): “…were positive for undetermined flaviviruses and 2/104were positive for tick-borne encephalitis virus (TBEV)”

Point 6: Lines 53-54: change to “…. [6], including the Mediterranean region [7], affecting

Response 6: As requested, the sentence has been changed (lines 53-54) to:” … is currently recognized as one of the most widely circulating and prevalent encephalitic flaviviruses [6], including the Mediterranean region [7], affecting mainly humans …”

Point 7:  Lines 55-58: Break up the sentence running over four lines.

Response 7: As suggested, the sentence was split into two sentences keeping the original meaning (Line 55): “In North Africa, WNV has been circulating actively for decades. Indeed, the geographical position along migratory bird routes, coupled with a climate and environment, favorable to Culex vectors (Cx. pipiens and Cx. Perexiguus), has caused recurrent and unpredictable reemergences of this virus. [10–14]

Point 8: Line 65: correct to “central parts of Tunisia”

Response 8: Line 65 has been corrected as suggested.

Point 9: Line 73: correct to Usutu virus (USUV( was first ….”

Response 9: Line 73 has been corrected as suggested.

Point 10: Line 101: Besides,…….

Response 10: Line 101 has been corrected as suggested.

Point 11:  Line 145: change to “mosquitoes. Identification of species was conducted using a binocular dissecting microscope ....”

Response 11: The section has been changed to (Line 145-148): “Traps were then transferred to the laboratory and kept in the freezer for 15 min to immobilize mosquitoes. Identification of species was conducted using a binocular dissecting microscope and then examined using the software for mosquitoes of Mediterranean Africa [65].”

Point 12: Figure 2: change to “Horse serum sampling localities” in the legend in the actual figure.

Response 12:

Changes were made in figure 2 as requested.

Point 13: Line 154: correct to “All studied horses were born ……”

Response 13: Line 156 has been corrected as suggested.

Point 14: Line 156: change to “Blood samples were …..

Response 14: Line 158 has been corrected as suggested.

Point 15: Line 162: correct to “Pools were ground in 500 uL….

Response 15: Line 164 has been corrected as suggested.

Point 16: Line 185: as a minimum provide the primers and probes in a supplementary table.

Response 16: In Fact, the list of primers and probes used is the same as that provided in the cited article (ref 66: Moutailler et al. 2019) and we think that the supplementary table would be a repetition of the cited article (which is a table of several pages). Thus, we don't think that this extra table is necessary since the designs are the same.

Point 17:  Line 203: correct to “as they did not have ….”

Response 17: Line 205 has been corrected as suggested.

Point 18: Line 208: correct to “The cELISA …”

Response 18: Line 210 has been corrected as suggested.

Point 19: Line 213: correct to “positive serum samples were…..”

Response 19: Line 215 has been corrected as suggested.

Point 20: Line 215: replace (2017) with [70]

Response 20: The reference has been rectified in line 217

Point 21: Line 241: provide percentage cell protection  - presumably 90% according to the reply to reviewer – but state it in the manuscript too. Also provide information about how cell toxicity was assessed, since that is subsequently mentioned in both the Results section and in the Discussion.

Response 21: in ligne 310 et 311 modifications requested were added according to the reviewer comment.

Point 22: Table 1: there is an extra b superscript.

Response 22: line 343 modifications to table 1 and to the text was added to ensure consistency on indications "b" and "c".

Point 23: Lines 263-5: sentence needs rephrasing in other to avoid the expression ‘among them’

Response 23: The sentence has been rephrased (Line 264): “The collection included 1,590 mosquitoes in Ichkeul, 80 mosquitoes in Moknine, 470 mosquitoes in Monastir and 340 mosquitoes in Msaken.”

Point 24: Lines 301 & 302: correct to “found to be” in both sentences

Response 24: Corrections have been made in both lines 303 and 304.

Point 25: Table 3: either remove the left side border or include a right side border in the table.

Response 25: The left side border was removed.

Point 26: Table 3: All the explanatory a, b, c, etc in the first column should be in superscript.

Response 26: All the explanatory a, b, c, etc in the first column were added in superscript.

Point 27: Table 4: for the cELISA it is not WNV that is detected but just flavivirus in general.

Response 27: WNV was changed to Flavivirus for the cELISA in Table 4.

Point 28: Lines 351-3: delete sentence starting with “of which 79 were …” as the same thing is said again below.                             

Response 28: The sentence was deleted.

Point 29: Lines 363-9: this is all recapitulation of data and should not be in the Discussion.

Response 29: All the recapitulation data sentences were deleted accordingly.

Point 30: Line 373: correct to “a closely related ….”

Response 30: Correction has been made in line 375.

Point 31: Lines 378-9: please provide a reference for the claim that the flavivirus cELISA has low specificity.

Response 31: The whole sentence was deleted as the discussion section was reorganised.

Point 32: The whole section in lines 376-406 is very disjointed and in need of editing/rewriting.

Response 32: The whole section was edited as requested.

Point 33: Line 412: correct to “capable of infecting ….”

Response 33: Correction has been made in line 413.

Point 34: Lines 413-415: sentence need rephrasing.

Response 34: The sentence was rephrased.

Reviewer 4 Report

The manuscript resubmitted by the authors has essentially been improved by the revision. There are only some minor notes and suggestions for changes to be made:

Line 97: please add that ixodid ticks (mainly of the genus Ixodes) are the vectors of TBEV.

Part 2.6 and 3.2: When I understand the response of the authors correctly, the realtime PCR of RBP was performed to check for a systemic infections in the mosquitoes and not for validation of the BioMark™ real-time PCR system. To my opinion, the description in the manuscript is still misleading, and the authors should rewrite the parts 2.6 and 3.2 to clarify this point

Line 215: please use the correct citation format for Vanhomwegen et al. (2017).

Table1, fourth column: please also add “b” to the other lines related to C. perexiguus.

Line 342: please add a percentage for the infection rates obtained in study [79].

Lines 423-427: Please rephrase. I guess that the authors want to say that because RBP was not analyzed there was just a pure detection of virus ingestion via the blood meal but no confirmation of a disseminated infection in the mosquitos and thus a vector function.

Author Response

Response to Reviewer 4 Comments

Point 1: The manuscript resubmitted by the authors has essentially been improved by the revision. There are only some minor notes and suggestions for changes to be made.

Response 1: Thank you for your considerable revision. Certainly, the manuscript has been improved thanks to your constructive comments and your valuable suggestions

Point 2: Line 97: please add that ixodid ticks (mainly of the genus Ixodes) are the vectors of TBEV.

Response 2: As suggested, a sentence has been added to the manuscript: line 98-99 “Hard ticks (Ixodidea) are the main vectors of TBEV, particularly those of the genus Ixodes [52]” 

Point 3: Part 2.6 and 3.2: When I understand the response of the authors correctly, the realtime PCR of RBP was performed to check for a systemic infections in the mosquitoes and not for validation of the BioMark™ real-time PCR system. To my opinion, the description in the manuscript is still misleading, and the authors should rewrite the parts 2.6 and 3.2 to clarify this point

Response 3: Yes it’s exactly what we aimed to. Many thanks for the comment, both titles  were changed according to the request.

Point 4: Line 215: please use the correct citation format for Vanhomwegen et al. (2017)

Response 4: The format of the citation “Vanhomwegen et al. (2017)“ has been corrected (Line 215).

Point 5: Table1, fourth column: please also add “b” to the other lines related to C. perexiguus.

Response 5: Changes has been made as requested.

Point 6: Line 342: please add a percentage for the infection rates obtained in study [79].

Response 6: The overall infection was added and between the different investigated regions.

Point 7:  Lines 423-427: Please rephrase. I guess that the authors want to say that because RBP was not analyzed there was just a pure detection of virus ingestion via the blood meal but no confirmation of a disseminated infection in the mosquitos and thus a vector function.

Response 7: The sentences were rephrased according to the recommendation.

Round 3

Reviewer 2 Report

The authors are commended for addressing this reviewer's comments and concerns. Apart from some English language editing required, the manuscript now reads fairly well.